# Multiobjective optimization identifies cancer-selective combination therapies

Otto I. Pulkkinen[1,2,3,4]*, Prson Gautam[1], Ville Mustonen[2,5]ʘ*, Tero Aittokallio[1,4,6,7]ʘ*

**1** Institute for Molecular Medicine Finland (FIMM), University of Helsinki, Helsinki, Finland, **2** Helsinki Institute for Information Technology (HIIT), Department of Computer Science, University of Helsinki, Helsinki, Finland, **3** Computational Physics Laboratory, Tampere University, Tampere, Finland, **4** Department of Mathematics and Statistics, University of Turku, Turku, Finland, **5** Organismal and Evolutionary Biology Research Programme, Institute of Biotechnology, University of Helsinki, Helsinki, Finland, **6** Institute for Cancer Research, Department of Cancer Genetics, Oslo University Hospital, Oslo, Norway, **7** Oslo Centre for Biostatistics and Epidemiology (OCBE), Faculty of Medicine, University of Oslo, Oslo, Norway

ʘ These authors contributed equally to this work.
\* otto.i.pulkkinen@gmail.com (OIP); v.mustonen@helsinki.fi (VM); tero.aittokallio@helsinki.fi (TA)

**Data Availability Statement:** The code and in-house measurement data are available at https://github.com/pulkkinen/combimop.

**Funding:** This work was supported by the Academy of Finland [grants 292611, 310507,

## Abstract

Combinatorial therapies are required to treat patients with advanced cancers that have become resistant to monotherapies through rewiring of redundant pathways. Due to a massive number of potential drug combinations, there is a need for systematic approaches to identify safe and effective combinations for each patient, using cost-effective methods. Here, we developed an exact multiobjective optimization method for identifying pairwise or higher-order combinations that show maximal cancer-selectivity. The prioritization of patient-specific combinations is based on Pareto-optimization in the search space spanned by the therapeutic and nonselective effects of combinations. We demonstrate the performance of the method in the context of BRAF-V600E melanoma treatment, where the optimal solutions predicted a number of co-inhibition partners for vemurafenib, a selective BRAF-V600E inhibitor, approved for advanced melanoma. We experimentally validated many of the predictions in BRAF-V600E melanoma cell line, and the results suggest that one can improve selective inhibition of BRAF-V600E melanoma cells by combinatorial targeting of MAPK/ERK and other compensatory pathways using pairwise and third-order drug combinations. Our mechanism-agnostic optimization method is widely applicable to various cancer types, and it takes as input only measurements of a subset of pairwise drug combinations, without requiring target information or genomic profiles. Such data-driven approaches may become useful for functional precision oncology applications that go beyond the cancer genetic dependency paradigm to optimize cancer-selective combinatorial treatments.

## Author summary

Cancer is diagnosed in nearly 40% of people in the U.S at some point during their lifetimes. Despite decades of research to lower cancer incidence and mortality, cancer

313267, 326238 to TA; grant 313270 to VM], the Cancer Society of Finland [TA] and the Sigrid Jusélius Foundation [TA]. The funders had no role in study design, data collection and analysis, decision to publish, or preparation of the manuscript.

**Competing interests:** The authors have declared that no competing interests exist.

remains a leading cause of deaths worldwide. Therefore, new targeted therapies are required to further reduce the death rates and toxic effects of treatments. Here we developed a mathematical optimization framework for finding cancer-selective treatments that optimally use drugs and their combinations. The method uses multiobjective optimization to identify drug combinations that simultaneously show maximal therapeutic potential and minimal non-selectivity, to avoid severe side effects. Our systematic search approach is applicable to various cancer types and it enables optimization of combinations involving both targeted therapies as well as standard chemotherapies.

## Introduction

Combination therapies have become the standard of care to treat many complex diseases, including advanced cancers, which have developed resistance to monotherapies. Due to the heterogeneity of the disease, there is an increasing need to identify targeted combinatorial therapies for each cancer patient individually based on patient-derived cell models, such as ex-vivo cell cultures or patient-derived organoids and xenograft models. Systematic approaches to finding new combinatorial treatments are often based on high-throughput screening of pre-clinical models that emphasize combination efficacy and/or synergy as the key determinants of the drug combination performance [1–4]. A combination is called synergistic if it has a higher than expected effect on cancer cell inhibition, where the expected combination effect is calculated based on monotherapy responses and a reference model, such as the highest single agent, Loewe additivity or Bliss independence [5].

The mechanistic basis of therapeutically beneficial combinations remain still poorly understood, but recent findings in cancer cell line screens suggest that targets of highly synergistic drug combinations are likely to interact at the level of signaling pathways, either targeting the same or functionally opposite pathways, in such a way that the combined effect is greater than the sum of the individual effects [6]. This can occur for example by combinatorial inhibition of two orthogonal pathways or by targeting compensatory mechanisms via pathway cross-talks [7]. However, the same combination mechanism may affect not only the cancerous cells, but manifest itself also as unintended adverse effects in healthy cells [8]. In this case, the combination is nonselective against the particular cancer cells, and may result in a degree of toxic effects as well. Therefore, combination synergy and expected efficacy alone are not sufficient determinants when designing safe and effective treatment options.

Identifying cancer-selective drug combinations amounts to maximizing combination efficacy in the cancer cells, while minimizing potential toxic effects to healthy cells. Such drug combination design can be considered as a multiobjective optimization problem (MOP), in which the two goals are pursued simultaneously [9, 10]. Even if multiobjective optimization has been used as a tool in planning radiotherapies [11, 12], to our knowledge, the study of Matlock et al. [13] is the only attempt to apply the mathematical optimization framework to designing drug combinations. They proposed numerical algorithms that converge to most effective and least toxic combinations of drug targets based on so-called target inhibition maps [14]. Even though such target-centric approach provides clues about the targets to inhibit or avoid in a given cell context, it cannot optimize the drugs to be used in the combination, and is not applicable to non-targeted therapies, such as standard chemotherapies or emerging immunotherapies.

In this work, we develop a novel drug-centric mathematical optimization framework for maximizing drug combination efficacy and minimizing its nonselectivity against a given

cancer type. Nonselectivity is defined through the average effect of a drug compound across a large number of cells from various cancer types, and it is used as a surrogate of toxic effects. This definition also makes it possible to apply the framework using drug sensitivity measurements in the cancer cells alone, without requiring measurements on healthy cells, nor information on targets of the drugs or genomic profiles of the cancer cells. To exemplify the framework, we use pairwise combination measurements of 104 drug compounds in 60 cancer cell lines from the NCI-ALMANAC resource [15], and subsequently experimentally validate the most interesting predictions in BRAF-V600E melanoma cell line. We propose a new quantity of selective synergy, Bliss multiexcess, a multiobjective generalization of the traditional Bliss excess measure.

## Results

### Joint modeling of therapeutic and nonselective effects

The reduction in the rate of growth of a cell line by a drug compound is typically measured as a growth fraction $Q$, the relative number of live cells with respect to untreated control after a certain time (e.g., inhibition % after 72h of treatment). *The therapeutic effect $E$ of a drug $i$ at concentration $c_i$, or a combination of a pair of drugs $i, j$ at concentrations $c_i, c_j$, on a target cell line $l$ is here defined as the negative logarithm of growth fraction $Q$,*

$$E_i(c_i; l) = -\log Q_i(c_i; l) \tag{1}$$

$$E_{ij}(c_i, c_j; l) = -\log Q_{ij}(c_i, c_j; l) . \tag{2}$$

Hence, the higher the therapeutic effect, the less viable the cells are after the treatment. One advantage of the logarithmic formulation is that the effect is additive, in that the predicted effect of two drugs acting independently, i.e. without interaction in the sense of Bliss independence [16], is given by the sum $E_i + E_j$ [17, 18]. We assume that higher order combination measurements are not available, so all modeling needs to rely on the measured effects of single drugs and pairs of drugs [19, 20]. To this end, we define the therapeutic effect of a combination $\boldsymbol{c} = (c_i)_{i \in \mathcal{D}}$ from a set $\mathcal{D}$ of $N$ possible drugs by the following pair interaction model (also known as regression model [21, 22] due to its original use in regression of drug interaction parameters [23]):

$$E(\boldsymbol{c}; l) = \underbrace{\sum_{i=1}^{N} E_i(c_i; l)}_{=E_B(\boldsymbol{c}; l)} + \underbrace{\sum_{j=1}^{N-1} \sum_{k=j+1}^{N} E_{j,k}^{XS}(c_j, c_k; l)}_{=E_B^{XS}(\boldsymbol{c}; l)} , \tag{3}$$

where $E_{ij}^{XS}(c_i, c_j; l) = E_{ij}(c_i, c_j; l) - E_i(c_i; l) - E_j(c_j; l)$ is the effect excess of the pair over Bliss independence model, or *Bliss excess* for short. The first sum in (3) yields the Bliss model effect $E_B(\boldsymbol{c}; l)$ of the combination $\boldsymbol{c}$, and the sum over pairs of drugs is its Bliss excess $E_B^{XS}(\boldsymbol{c}; l)$.

Let $m(\boldsymbol{c}) = |\{i: c_i > 0\}|$ denote the number of components in a drug combination $\boldsymbol{c}$. We then have three different types of equations for $m(\boldsymbol{c}) = 1$, $m(\boldsymbol{c}) = 2$, and $m(\boldsymbol{c}) \geq 2$, as shown in the following subsections.

**1. Monotherapy, $m(\boldsymbol{c}) = 1$.**  For a monotherapy, $c_{i_0} = c$ for a drug $i_0$, and $c_i = 0$ otherwise. The therapeutic effect in Eq (3) equals the effect of the single compound $i_0$ at concentration $c$, i.e.

$$E(\boldsymbol{c}; l) = E_{i_0}(c; l) . \tag{4}$$

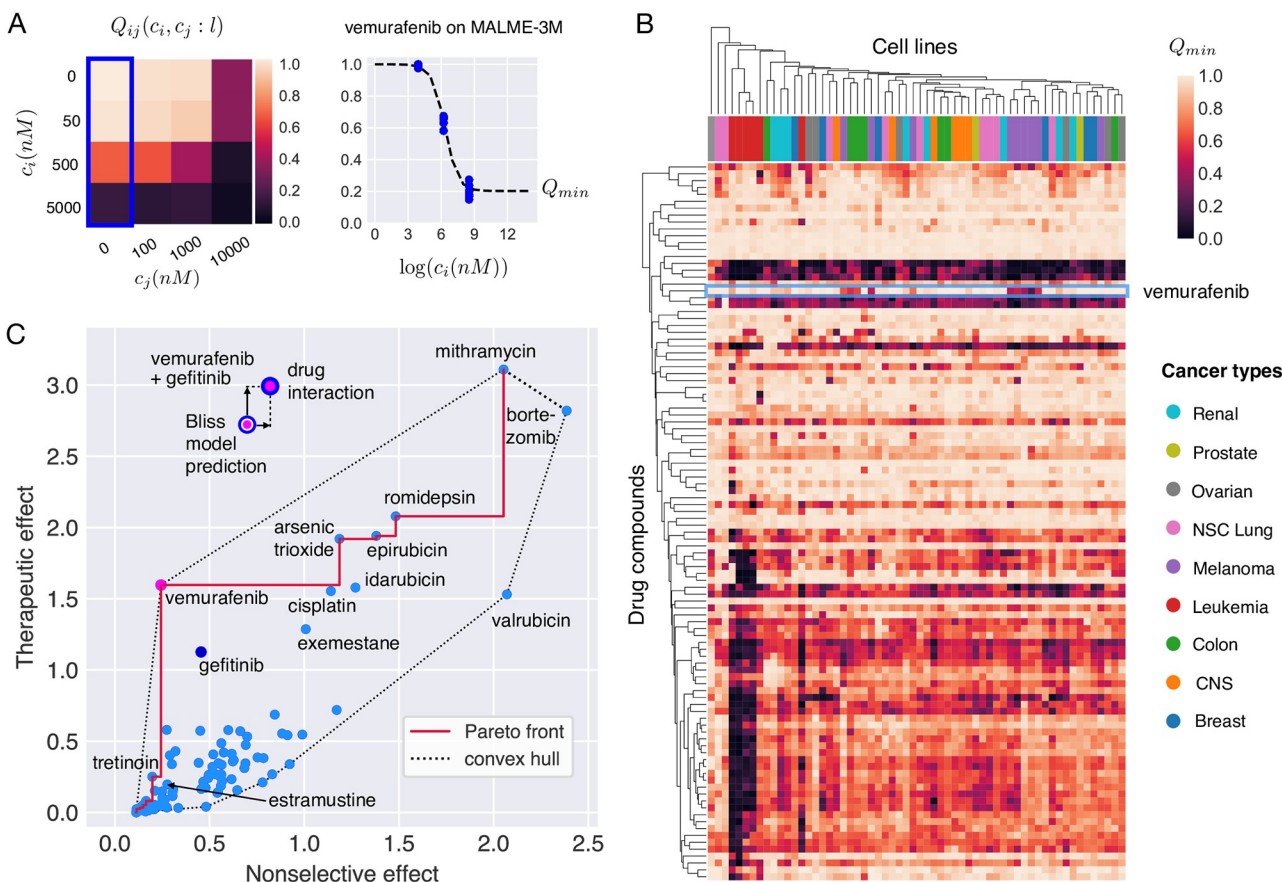

**Fig 1. Selectivity analysis of the 104 individual drug compounds in NCI ALMANAC. (A)** Data consists of drug sensitivity measurements of the NCI-60 cell lines at various concentrations $c_i$, $c_j$ of drug pairs (here vemurafenib and gefitinib, respectively). Maximal single compound activity, $Q_{min}$, of a drug is obtained by fitting Hill curves to the dose-response data. **(B)** Hierarchical clustering of cell lines and drug compounds according to their $Q_{min}$. Highly effective but nonselective drugs are clustered together and show up as dark horizontal bands in the heatmap. Leukemias (red color columns) cluster strongly together as they are more vulnerable to a large variety of treatments than the other cancer types. Melanomas (violet columns) form one major cluster, and the rest of the melanoma cell lines are associated with other cancer types. In particular, all but one melanoma cell line stand out from most of the other cell lines because of their susceptibility to vemurafenib (light blue box). The drug activity profile of vemurafenib is clustered together most closely with estramustine, an estradiol derivative with a nitrogen mustard moiety, and the anthracycline neoplastics epirubicin and idarubicin. **(C)** Multiobjective optimization of single drug compounds (*i.e.* $M = 1$) acting on Malme-3M cell line. Vemurafenib is clearly the optimal single-compound treatment because it is both effective and selective. Bortezomib and valrubicin, near the rightmost tip of the convex hull, are the least optimal drugs. Mithramycin leads to highest therapeutic effect but it is also highly nonselective, far more than the common cytotoxic treatment cisplatin. EGFR inhibitor gefitinib alone is not a Pareto optimal solution. The multi-coloured dots show that a substantially better efficacy can be achieved by combining drugs. For the combination of vemurafenib and gefitinib, a higher therapeutic effect than predicted by Bliss independence model emerges, yet the boost in efficacy comes with an increase in nonselective effect as well.

The right-hand side equals the experimental value of $E$ obtained from the measured growth inhibition if data for drug $i_0$ at concentration $c$ exists. Otherwise, it can be obtained from experimental results by fitting a Hill curve [24] to the other, measured concentrations (see Fig 1 and S1 Text).

**2. Two-drug combination, $m(c) = 2$.** For a combination consisting of a pair of drugs, we have $c_{i_1} = c_1$ and $c_{i_2} = c_2$ for some drugs $i_1$ and $i_2$, and zero otherwise. Eq (3) then reads

$$E(\boldsymbol{c}; l) = E_{i_1}(c_1; l) + E_{i_2}(c_2; l) + E_{i_1, i_2}^{XS}(c_1, c_2; l) . \tag{5}$$

Emergence of a Bliss excess term quantifying drug-drug interaction is a notable difference to the corresponding formula (4) for a monotherapy. However, also in this case the right-hand

side exactly matches the experimental value obtained from measured growth inhibition if the combinatorial growth inhibition of drug pair $(i_1, i_2)$ was measured at $(c_1, c_2)$. We note, however, that typically much fewer data points exist for the interaction part than for monotherapies, and the modeling of the combination effect is therefore more sensitive to measurement errors.

**3. Drug triplet or higher order combination, $m(c) \geq 3$.** No large-scale datasets covering majority of all combinations with at least three drug components currently exist. Therefore, model (3) of the combination effect is based on measured pairwise effects, and the predicted effect is necessarily an approximation,

$$E(\boldsymbol{c}; l) \approx \sum_{i=1}^{N} E_i(c_i; l) + \sum_{j=1}^{N-1} \sum_{k=j+1}^{N} E_{j,k}^{XS}(c_j, c_k; l) . \tag{6}$$

For example, in the case $m(c) = 3$,

$$E(\boldsymbol{c}; l) \approx E_{i_1}(c_1; l) + E_{i_2}(c_2; l) + E_{i_3}(c_3; l) + E_{i_1,i_2}^{XS}(c_1, c_2; l) + E_{i_1,i_3}^{XS}(c_1, c_3; l) + E_{i_2,i_3}^{XS}(c_2, c_3; l) . \tag{7}$$

The number of terms increases rapidly with an increasing $m(c)$ because an interaction term with all other drugs is introduced upon addition of a new drug component to a combination. Accordingly, the effect becomes more sensitive to measurement errors.

As the level of abstraction in the modeling of the therapeutic effect changes with the number of drug components $m(\boldsymbol{c})$, the three cases will also be discussed separately later in this article.

## Nonselective effect and multiobjective optimization

We model the nonselectivity of a drug compound by the mean effect over the set of all cell lines $\mathcal{L}$ in the data,

$$\bar{E}(\boldsymbol{c}) = \frac{1}{|\mathcal{L}|} \sum_{l \in \mathcal{L}} E(\boldsymbol{c}; l) . \tag{8}$$

For a large number of patient-derived samples from various tissue types, the average in Eq (8) comprises many somatically different cell lines. Hence, we can consider it as a proxy for the degree of nonselective toxic effects on an average (cancerous or non-cancerous) tissue. Indeed, it is known that widely cytotoxic (high mean effect over all the cell lines) chemotherapy drugs display a range of severe side effects due to their poor selectivity for cancerous tissue over normal tissue [25]. On the other hand, hormonal therapies and drugs targeting cellular signaling often have smaller mean effect, and their adverse effects are also typically more manageable. In the S1 Appendix, we provide support to the assumption that the mean effect is a good proxy for nonselective toxic effects, and even clinical adverse effects, therefore serving as a gauge for combination safety.

As a drug combination should not benefit from additional drugs with hardly no effect, we add to each $\bar{E}_i$ a small regularizing parameter $\delta$ (we use the value 0.1) that counterbalances any arbitrarily small increase in therapeutic effect or decrease in the mean effect. We hence define the *nonselective effect* of a drug combination with $m(\boldsymbol{c})$ components as

$$\bar{E}_\delta(\boldsymbol{c}) = \delta \cdot m(\boldsymbol{c}) + \bar{E}(\boldsymbol{c}) . \tag{9}$$

In the spirit of Eq (3), the nonselective effect too can be split into nonselective effects $\bar{E}_i + \delta$ of isolated compounds, and mean excesses $\bar{E}_{ij}^{XS}$, which stem from the drug interactions.

As the nonselective effect acts as a surrogate for adverse effects and toxicity of a combination, it makes sense to ask, which drugs or their combinations have most therapeutic potential conditioning on a tolerance to nonselective effect. We divide the problem into parts of increasing complexity by restricting the number of drug components in combinations: We first study monotherapies only, then add in pairwise combinations, and finally allow for combinations of any order.

Let $M$ denote the maximal number of drugs allowed in any drug combination $\boldsymbol{c}$, in that $m(\boldsymbol{c}) \leq M$. To find optimal combinations, with at most $M$ components, for a cancer cell line $l$ that simultaneously maximize the therapeutic effect and minimize the nonselective effect is a multi-objective optimization problem (MOP) [9, 10]. Formulated using the so-called $\epsilon$-constraint method [9, 26], the task is to find the set of concentration vectors

$$C(M; l) = \bigcup_{\Theta \in \mathbb{R}_+} \{\operatorname{argmax} E(\boldsymbol{c}; l) : \bar{E}_\delta(\boldsymbol{c}) \leq \Theta, m(\boldsymbol{c}) \leq M\}. \tag{10}$$

Here, $\Theta$ denotes the tolerance to nonselective effect, and a vector of drug concentrations that maximizes the therapeutic effect corresponds to each $\Theta$. The set $C$ of solutions of this continuous MOP traces, for each $M$, a *Pareto front*, a curve in the plane of nonselective and therapeutic effects, the points of which are called *Pareto optimal*. For each point on the Pareto front, no combination of drug concentrations with lower nonselective effect and higher therapeutic effect exists when $M$ is fixed.

## Dimensional reduction by the strongest effect of a combination

The fact that the concentrations $\boldsymbol{c}$ can be adjusted on a continuous scale means that the number of solutions at the Pareto front is infinite. The Pareto front is therefore a continuous curve in the plane of therapeutic and nonselective effects. Solving this continuum MOP is obviously a numerically demanding task, and a successful solution very much depends on availability of reliable combination measurements at a multitude of concentrations of all drug compounds, so that a faithful continuous approximation to the effect in the plane of concentrations can be found for each pair of drugs. To lower the risk of false predictions due to measurement error and/or insufficient data resolution, we inspect a discrete set of concentrations instead. The two-drug combination effects are typically quantified by $n$-by-$n$ square matrices (*e.g.* $n = 4, 8$, see Fig 1) with $n - 1$ nonzero concentrations for both drugs in a combination. However, these concentrations are not necessarily the same for a single drug in each of those combination measurements, which complicates the analysis. Here we choose to reduce the problem further: We use binary variables $N_i = 1$ or $0$, to determine whether a drug compound $i \in \mathcal{D}$ is present in the combination or not, respectively. The effect of the whole drug combination is then predicted by the pair interaction model to be

$$E(\boldsymbol{c}; l) = \sum_{j=1}^{N} \varepsilon_i(l) N_i + \sum_{j=1}^{N-1} \sum_{k=j+1}^{N} \varepsilon_{jk}^{XS}(l) N_j N_k \ , \tag{11}$$

where

$$\varepsilon_i(l) = \max_{c_i > 0} E_i(c_i; l) \tag{12}$$

is the strongest effect of drug $i$ computed from a fitted Hill curve to growth fractions $Q_i$ (see Fig 1 and S1 Text), and, with $(c_i^*, c_j^*) = \operatorname{argmax}_{c_i, c_j > 0} E_{ij}(c_i, c_j; l)$,

$$\varepsilon_{ij}^{XS}(l) = E_{ij}(c_i^*, c_j^*; l) - E_{ij}(c_i^*, 0; l) - E_{ij}(0, c_j^*; l) \tag{13}$$

is the Bliss excess at the maximal measured therapeutic effect of the combination $i, j$. This way of defining the parameters of the binary problem was found to constitute a robust model. In particular, as high-throughput drug screens often show variability within and between plates on which the cells are drugged, the strongest effect from a fitted Hill curve for each cell line yields a reliable single number for the therapeutic effect of a drug compound in monotherapy. Furthermore, the way the pairwise interaction constants $\varepsilon_{ij}^{XS}$ are computed per matrix from measurements on the same plate, and not against the fitted Hill curves, respects plate-to-plate variance and significantly reduces the the risk of misestimating the drug interaction strengths.

In the binary model, the nonselective effect is again given by Eq (9) with the mean effect over all cell lines computed using the binary effect function. The notions of Bliss independence and Bliss excess of the whole combination carry over to the new definition in a natural way.

A consequence of dimensional reduction through Eqs (11–13) is that the concentration of a drug in combination with another drug gets fixed, and a different concentration may get chosen for the same drug in another pairwise combination. The combination triplet spanned by these two pairs would then paradoxically contain a single drug in two different concentrations, which may lead to overestimation of the combinatorial therapeutic effect. However, should this occur, the nonselective effect is likewise affected, and the MOP is stable against these changes.

We consider our solution to MOP *a posteriori* method [9], in which a human decision maker decides, which of the Pareto optimal points provided by the algorithm are clinically relevant and worth testing further in the laboratory. For example, the combinations taken into consideration should not be much more nonselective than broadly toxic standard chemotherapies. Even if this condition is controlled by the nonselective effect, the set of Pareto optimal or nearly optimal combinations may become large. Therefore ways of further delineating the benefits achieved by a particular combination in comparison to other Pareto optimal solutions are needed.

## Therapeutic dominance

In multiobjective optimization, a vector is said to dominate another vector if it performs at least equally well as the other one in all features, *and* it outperforms the other vector at least in one feature [27]. Inspired by this concept, we define the therapeutic dominance of a Pareto optimal combination:

Let $E(\mathbf{c}^*)$ and $\bar{E}_\delta(\mathbf{c}^*)$ denote the therapeutic effect and the nonselective effect of a Pareto optimal combination $\mathbf{c}^*$, respectively, and let $m(\mathbf{c}^*)$ denote its number of components. The *therapeutic dominance* of $\mathbf{c}^*$ is defined as the difference of its therapeutic effect $E(\mathbf{c}^*)$ and the highest therapeutic effect of all Pareto optimal combinations with at most $m(\mathbf{c}^*) - 1$ components and nonselective effect not exceeding $\bar{E}_\delta(\mathbf{c}^*)$. This directly quantifies the therapeutic advance gained by adding a drug to a Pareto optimal combination, or achieved by a monotherapy in comparison to no therapy at all.

## Selective synergy

A drug combination can be Pareto optimal for several reasons. First, the individual therapeutic effects of the drug components may be strong enough to bring the combination to the Pareto front without any synergistic interaction. In this case, the Bliss excess $E_B^{XS}(\mathbf{c}; l)$ of the combination is small. Second, the combination may exhibit a strong effect due to synergistic interaction of the drugs. However, synergism occurs in a large number of cell lines, leading to a comparable but not excessive increase in nonselective effect. This scenario is characterized by a higher-than-average Bliss excess, and a mean Bliss excess $\bar{E}_B^{XS}(\mathbf{c})$ of the same order. Third, the

combination may show high *selective synergy*, in that the drug interaction is manifested as an increase in therapeutic effect, and only little, or even as a negative change, in the mean effect over cell lines. Selective synergy is evidenced by high Bliss excess and low mean Bliss excess.

The degree of selective synergy of a combination can be boiled down to a single number, *Bliss multiexcess*

$$\mathcal{E}_B^{XS}(\boldsymbol{c}, \phi; l) = E_B^{XS}(\boldsymbol{c}; l) - \phi \bar{E}_B^{XS}(\boldsymbol{c}),\tag{14}$$

where $\phi$ weighs the importance of adverse effects. A motivation for definition (14), and a systematic way of choosing the value of $\phi$ comes from the scalarized (*i.e.* projected to one dimension), convex optimization problem corresponding to our MOP: Bliss multiexcess (14) equals the interaction part of its objective function [27]

$$\mathcal{E}(\boldsymbol{c}, \phi; l) = E(\boldsymbol{c}; l) - \phi \bar{E}_\delta(\boldsymbol{c}),\tag{15}$$

which is to be maximized. Thus the combinations that are scalar-optimal for synergistic reasons have high Bliss multiexcess. This argument can be extended to combinations outside the convex hull by appropriately choosing the weighing parameter $\phi$. As the Pareto optimal combinations near the convex hull can be seen as optional solutions to the scalarized problem, we use a single value of $\phi$ for all Pareto optimal combinations between two convex solutions—the smallest $\phi$ that yields the higher effect combination as an optimal solution of the scalarized problem.

## Application to melanoma

We applied the optimization framework to designing new combination therapies for malignant melanoma. The optimization is based on the sensitivity data of 104 drug compounds and their pairwise combinations in the NCI-60 cell lines available in the NCI ALMANAC resource [28]. In this data, the drug-induced growth inhibition is tested in four-by-four matrices, as depicted in Fig 1A. Hierarchical clustering of drug compounds and the cell lines based on monotherapy sensitivities reveals high variation across the cell lines (Fig 1B). In particular, vemurafenib shows a strong effect in most of the melanoma cell lines, whereas in other cancer types the effect is less potent. This is expected, as vemurafenib is a selective inhibitor of V600E-mutant BRAF, and this mutation is present in approximately 60% of melanomas. Vemurafenib has become one of the main treatments for late-stage melanoma since its discovery in 2008 and consequent FDA approval in 2011 [29–31]. Despite the initial efficacy of vemurafenib in many patients, the prognoses still remain poor as many patients develop resistance to vemurafenib, often within 6 to 8 months from therapy initiation, due to reactivation of the MAPK/ERK and other pathways. This has led to interest in combinatorial inhibition of the MAPK/ERK pathway components.

Fig 1C shows the solution of the MOP for all 104 drug compounds in the MALME-3M melanoma cell line with the BRAF-V600E mutation. The Pareto front traces out the optimal combinations in the plane of nonselective versus therapeutic effects. Vemurafenib stands out as a highly selective and effective treatment, as expected. Vitamin A derivative tretinoin has a lower nonselective effect than vemurafenib, but its therapeutic effect is much smaller. On the other hand, gefitinib, an EGFR inhibitor upstream of BRAF in the MAPK/ERK signaling pathway, has slightly smaller therapeutic effect than vemurafenib but it is more nonselective. Notably, the combination of vemurafenib and gefitinib (Fig 1C, top) shows an equally strong therapeutic effect as the most potent monotherapy, mithramycin (plicamycin), and the combination is synergistic, in that the effect is higher than that predicted by the Bliss independence model.

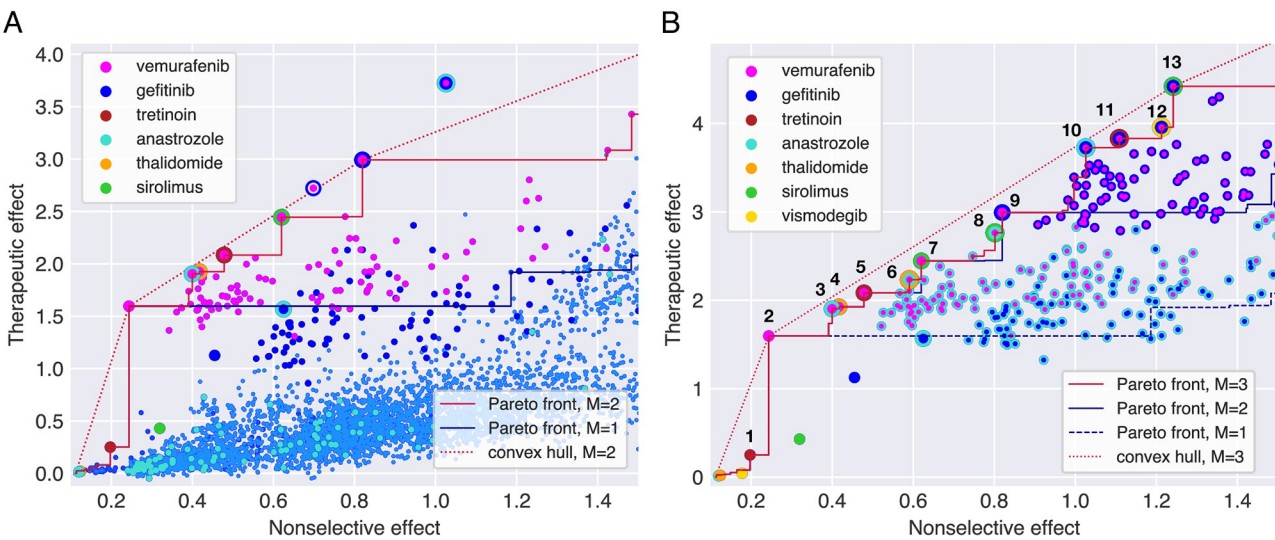

**Fig 2.** **(A)** Multiobjective optimization (MOP) solutions for monotherapies and two-drug combinations ($M = 2$). Monotherapies are represented by big monochrome dots, and multicoloured dots indicate combinations of selected compounds (see color legend). The small light blue dots represent all the other drugs alone and in two-drug combinations. Anastrozole alone has next to no effect on any cell line, yet in combination with vemurafenib and gefitinib, a selective effect emerges. The triplet combination of these three drugs is shown for comparison, and it yields a significant advance over all pairs of the drugs. Pareto fronts for $M = 1, 2$ are shown. **(B)** MOP for drug triplets and their subcombinations ($M = 3$). Due to high number of possible combinations, only monotherapies of drugs indicated in the legend and drug combinations involving vemurafenib, gefitinib and anastrozole are shown. The triplet of vemurafenib and gefitinib with sirolimus is on the convex hull, indicating a strong selective effect, yet it comes with a significant cost in nonselective effect. The Pareto front for $M = 4$ coincides with the one for triplets within this (and actually much wider) window of nonselective effects, indicating that none of the fourth-order combinations perform better than drug triplets and their subcombinations.

The solutions of the MOPs for combinations of up to two drugs (Fig 2A, S1 Table) and higher-order combinations (Fig 2B, Table 1 and S2 Table) confirm that vemurafenib-gefitinib combination is Pareto optimal regardless of the number of components allowed in combinations. Other Pareto optimal combinations involve vemurafenib with anastrozole, tretinoin,

**Table 1. Selected Pareto optimal combinations for MALME-3M (See S2 Table for the complete list).**

| Monotherapies |
| --- |
| **1**. tretinoin |
| **2**. vemurafenib |
| **Two-drug combinations**: |
| **3**. anastrozole + vemurafenib |
| **4**. thalidomide + vemurafenib |
| **5**. tretinoin + vemurafenib |
| **7**. sirolimus + vemurafenib |
| **9**. gefitinib + vemurafenib |
| **Drug triplets**: |
| **6**.anastrozole + thalidomide + vemurafenib |
| **8**. anastrozole + sirolimus + vemurafenib |
| **10** anastrozole + gefitinib + vemurafenib |
| **11** gefitinib + tretinoin + vemurafenib, |
| **12**. gefitinib + vemurafenib + vismodegib |
| **13**. gefitinib + sirolimus + vemurafenib |

The numbering of combinations is the same as in Figs 2B and 3.

   

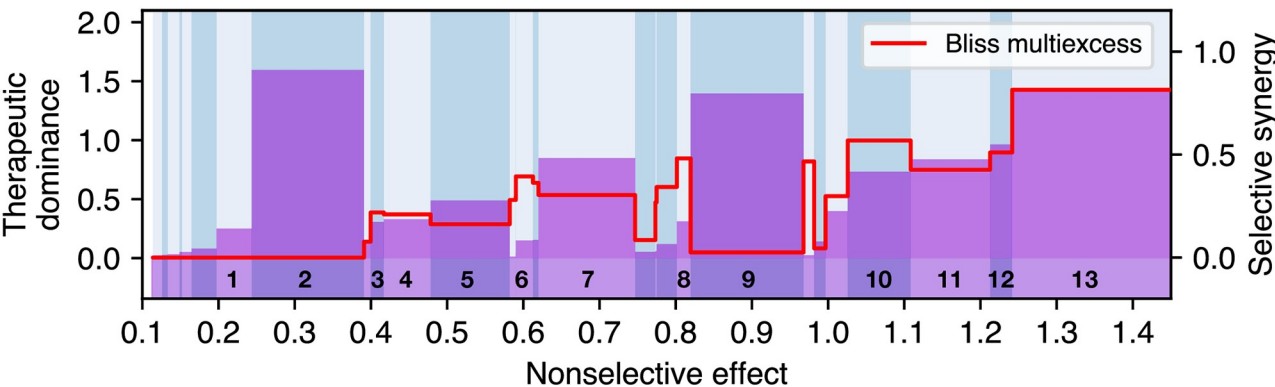

**Fig 3. Therapeutic dominance (bars, left y-axis) and selective synergy measured by the Bliss multiexcess (red curve, right y-axis) of Pareto optimal combinations as a function of nonselective effect.** The numbered combinations correspond to Pareto optimal combinations highlighted in Fig 2B, from tretinoin (1) monotherapy to vemurarfenib+gefitinib+sirolimus (13) triplet. Vemurafenib monotherapy (2) is dominant at low mean effect. Vemurafenib + gefitinib (9) duplet has high a dominance but very low selective synergy, indicating that the combination is optimal because of its strong therapeutic effect, not due to selective drug interaction. The combination vemurafenib+gefitinib+sirolimus (13) has the highest selective synergy among shown combinations. The combinations vemurafenib+anastrozole (3) and vemurafenib+gefitinib+anastrozole (10) exhibit both considerable dominance and local maxima in selective synergy.

thalidomide and sirolimus as partners. Of these, the anti-inflammatory drug thalidomide and the aromatase inhibitor anastrozole have little effect when used alone, indicating a true synergistic interaction with vemurafenib. The same compounds appear in the Pareto optimal combinations of three and higher order in Fig 2B. With a few exceptions, Pareto optimal combinations consist mostly of monotherapies and duplets at smaller therapeutic and nonselective effects than vemurafenib-gefitinib. A cloud of triplets that all include vemurafenib and gefitinib as a pairwise subcombination dominates at higher effects in Fig 2B.

Therapeutic dominance and the measure of selective synergy were used to further investigate the selected combinations (Fig 3). Vemurafenib monotherapy, along with vemurafenib-gefitinib and vemurafenib-gefitinib-sirolimus combinations are clearly the most dominant therapies, since they yield the most therapeutic advance in comparison to a lower order Pareto optimal combinations. However, such advance can be associated with a significant increase in nonselective effect in the case of combinations. For instance, the duplet of vemurafenib and gefitinib is synergistic, but not selectively synergistic. The vemurafenib duplets with anastrozole, thalidomide, tretinoin and sirolimus, and the triplets of vemurafenib, gefitinib and anastrozole, or sirolimus are truly selectively synergistic combinations with significant therapeutic dominance over their lower order combinations.

**In-house validations.** To further test the most interesting predictions from the MOP solution, we experimentally validated in-house two Pareto optimal drug triplets and their subcombinations in the BRAF-V600E melanoma cell line MALME-3M. This validation experiment was performed to test the accuracy of the predicted therapeutic effect in other experimental assay (FIMM) than that used for model predictions (NCI-60). The in-house drug combination assay was performed in 8x8 matrix format, similar to previous studies (25) (see Materials and methods for details and S1 Fig for an example).

Table 2 shows the maximal effect over the measured concentrations in both assays. In accord to predictions, the triplet combinations tested in-house ($E_{FIMM}$) show stronger effect than any of the duplets, and the observed therapeutic effects of triplets are very close to the predicted values ($E_{NCI}$). The vemurafenib monotherapy and its combinations with gefitinib had stronger effects in FIMM assay than those reported in NCI ALMANAC. This makes the Bliss excesses smaller in the in-house assays. However, as the concentration of the third component

**Table 2. NCI ALMANAC and FIMM drug assays on MALME-3M.**

| Treatment | $E_{\text{NCI}}$ | $E_{B,NCI}^{XS}$ | $E_{\text{FIMM}}$ | $E_{B,FIMM}^{XS}$ |
|---|---|---|---|---|
| vemurafenib | 1.60 | - | 2.10(10) | - |
| gefitinib | 1.13 | - | 1.39(6) | - |
| anastrozole | 0.02 | - | 0.02(1) | - |
| vismodegib | 0.04 | - | 0.03(1) | - |
| vem+gef | 2.99 | 0.26 | 3.57 | 0.08 |
| vem+ana | 1.91 | 0.29 | 2.33 | 0.21 |
| vem+vis | 1.53 | -0.11 | 2.41 | 0.28 |
| gef+ana | 1.57 | 0.42 | 1.73 | 0.32 |
| gef+vis | 2.13 | 0.96 | 2.15 | 0.73 |
| vem+gef+ana(1 μM) | 3.73 | 0.98 | 3.74 | 0.23 |
| vem+gef+vis(1 μM) | 3.89 | 1.12 | 3.73 | 0.21 |

$E_{\text{NCI}}$: Therapeutic effect predicted by the binary pair interaction model based on NCI ALMANAC data.

$E_{\text{FIMM}}$: Therapeutic effect measured at FIMM. The figure in parenthesis is the error of the last digit from fitting the Hill curve.

$E_B^{XS}$: Bliss excesses computed either from the first or the third column.

was fixed in the in-house assays, we lack the full three-dimensional optimization over concentrations, and hence the most synergistic vectors may have been missed.

As the pairwise matrices in our in-house experiments have higher resolution (8x8) than in the NCI ALMANAC assays (4x4), we were able to observe interesting nonlinearities in the growth inhibition patterns. For example, the combination of vemurafenib and gefitinib shows antagonistic pattern at high concentrations of vemurafenib and low concentrations of gefitinib, while it results in a synergistic effect at high concentrations of both components (see S2 Fig). Further, the experiments revealed a highly synergistic combination between gefitinib and vismodegib, a SMO inhibitor of the hedgehog pathway, and this combination scores high also in our MOP solution because of this synergy along the high additive effect and nonselective synergy of vemurafenib and gefitinib.

## Discussion

We introduced a multiobjective optimization framework for designing effective and safe drug combinations by simultaneous optimization of therapeutic and nonselective effects. The MOP approach requires only the measurements of single drugs and a subset of their pairwise combinations, and it enables optimization of also higher-order combinatorial treatments of three or more drugs that are often required for treating advanced cancers. The nonselectivity was modeled here by the mean effect over all the cell lines in the NCI ALMANAC data set, yet any other control data can be used for the purpose of estimating toxic side effects. The exact MOP solutions from Pareto-optimization provide the decision maker with a targeted set of patient-specific combinations, illustrated and quantified by their therapeutic and nonselective effects. We further defined Bliss multiexcess, a multiobjective generalization of the traditional Bliss excess, as a new measure of selective synergy, and demonstrated how it enables one to solve two critical challenges of combinatorial therapeutics: (i) combinatorial explosion of the number of potential higher-order combinations to consider for a given patient, and (ii) optimization of the balance between therapeutic and nonselective effects to rule out broadly toxic combinations.

Our results confirmed the previously suggested combinatorial inhibition of the MAPK/ERK pathway as an effective and safe treatment for malignant melanoma with BRAF V600E

mutation. However, we note that the approved combination of vemurafenib with cobimetinib [28, 29], or with any MEK inhibitor, was not tested in the NCI ALMANAC study, but our mechanistic-agnostic model pairs the EGFR inhibitor gefitinib with vemurafenib instead. Several higher order combinations were also found to be Pareto optimal. Of these, an aromatase inhibitor anastrozole is typically used to treat estrogen-responsive breast cancer, and it is therefore quite surprising to see it having selectivity for melanoma. On the other hand, already the unsupervised hierarchical clustering of drugs without any interaction information paired vemurafenib with estramustine, an estrogen receptor binder. There is increasing evidence that hormone levels play a role in the growth and progression of melanoma, although the underlying mechanisms are still unclear [30]. These examples demonstrate how our MOP solutions identify among a massive number of potential combinations those with optimal efficacy and safety profiles that have higher likelihood of success in further pre-clinical and clinical testing.

To select the Pareto optimal multi-drug combinations without relying on approximative modeling of combination efficacies, the current MOP framework requires as input the efficacies of a majority of combinations both in the target cell line and in a healthy control, or in a sufficiently large number of other cell lines, in order to compute the nonselective effects. More specifically, for finding the Pareto optimal drug combinations of at most $M$ components from a set of $N$ drug compounds, an order of $N^M$ of measurements are required for each cell line used in the modeling. Such large-scale drug screens currently exist for monotherapies and two-drug combinations ($M = 1, 2$), and to apply the MOP framework to higher-order combinations, it must be augmented with a model that predicts the higher-order combination efficacies. In the present work, we chose the pair interaction model that uses only the measured efficacies of monotherapies and two-drug combinations for this purpose, but there are also other alternatives, which may be based on pair combination efficacies [21, 22] or use additional chemical and biological information such as quantitative structure-activity relationship analysis of the compounds, gene expression data, or signaling pathway and network annotations [8, 32].

The prediction of the combination effects was based on Hill equations fitted to monotherapy dose-response data, and pairwise Bliss excesses, to increase the model robustness against measurement errors. Since the maximal effect $Q_{min}$ of a combination may be sensitive to measurement errors, alternative measures of combination efficacy, such as the mean effect over the concentration pairs, could be used instead. However, since the most promising Pareto optimal combinations require anyway further experimental validation over a finer grid of concentrations, any potential erroneous solutions based on outlier measurements can be then identified in this validation phase. We further note that, unless the concentration vectors of single drugs were fixed in all the dose-response measurements, even averaging of the pairwise effects over concentrations will not completely resolve the apparent paradox, where a drug may be present at two different concentrations in one combination. Full multiobjective optimization in the space of continuous concentrations will be needed to remove the superfluous degree of freedom, and developing for it a theoretical and numerical methodology that meets the challenges posed by both low data resolution and possible measurement errors requires further studies.

The multiobjective optimization framework presented in this article was designed with high-throughput experimentation in mind. Datasets such as the NCI ALMANAC, consisting of nearly 3 million data points, allow this kind of analysis. Even if datasets of this size are still rare, and can be produced only with larger resources, the multiobjective optimization method is also applicable to smaller-scale combinatorial datasets that can be produced in many laboratories with the help of robotic instruments, in particular if combined with pre-existing data from other studies. There area also public resources, such as PharmacoDB [33] and

SYNERGxDB [34], that integrate monotherapy and drug combination data from published studies in several cancer types, and can therefore be used in the MOP approach, either as target cell lines or as background data for toxicity modelling. Dose-reponses in normal cells can be also used to asses the nonselective toxicity of new drugs to be incorporated in the model and in the optimization, in those applications where such control cell lines are available.

The caveat of all the models that are based on cancer cell lines is that they are monoclonal and miss many factors important for in vivo drug responses, including tumor microenvironment. This is in contrast to real tumors, which comprise heterogeneous collections of cells with subclonal structure. Therefore, also the response to a therapy *in vivo* is a mixture of responses from the subclonal populations, bringing the total therapeutic effect closer to the nonselective effect. The tumors grow within special microenvironments that permit carcinogenesis and allow tumor maintenance. As a consequence, most effective combination therapies, *in vivo*, are probably multi-targeted and attack the cancer from multiple sides by slowing down or stopping tumor cell proliferation, by thwarting drug resistance, and by blocking the required support coming from the tumor microenvironment [35]. Nonselectivity may thus be advantageous in some cases. However, the complementary objectives of drug efficacy and safety are still valid. The multiobjective method can also be extended to drug profiling data of primary patient samples (*e.g.* blood cancers) or complex 3D models like organoids, co-cultures or even tissue slices, which contain clonal heterogeneity and tumor microenvironmental structure. In particular, when considering heterogenous cell population within tumors and tumor microenvironment one would need to design targeted drug combinations that co-inhibit only the malignant cells and avoid severe inhibition of healthy cells. Extending the MOP formulation to sub-clonal drug response data is one important future aim, once such data will become available in larger-scale.

In summary, we have introduced a novel conceptual approach for optimizing patient-specific combinatorial treatments, and demonstrated its potential both in published large-scale drug combination data as well as using in-house validation experiments. The approach is fast, does not require any genomic profiling of the patient cells, is applicable to various cancer types, and enables optimization of combinations involving both targeted therapies as well as standard chemotherapies. We hope that the systematic optimization framework will prove useful in designing new effective and safe cancer therapies to further reduce the death rates and the toxic effects of treatments.

## Materials and methods

### Cell line for in-house experiments

Melanoma cell line Malme-3M (ATCC HTB-64) was purchased from ATCC. It was maintained at 37˚C with 5% CO2 in a humidified incubator in IMDM (Cat#12440046) supplemented with 20% fetal bovine serum (Cat#10270-106) and 1% Penicillin-Streptomycin (Cat#15140-122). All the reagents were purchased from ThermoFisher Scientific for the in-house assays.

### Drug combination assays

The drug combination screening approach described previously [36] was adopted on MALME-3M cell line. Seven different concentrations in log3-fold dilution ranging from 10 nM to 10000 nM of vemurafenib and gefitinib, vemurafenib and anastrozole, and gefitinib and anastrozole were combined with each other in 8x8 matrix formats in duplicate. For the triple combination, 500 nM of vemurafenib or 500 nM gefitinib or 1000 nM of anastrozole was added to the alternate drug combination matrix (in one copy). Exactly as described above,

another set of combination was prepared for vemurafenib, gefitinib and vismodegib. For triple combination, 1000 nM of vismodegib was used whereas the concentrations of vemurafenib and gefitinib were the same as mentioned above (Supplementary file drug_combo_design. xlsx). The compounds were plated to black clear bottom 384-well plates (Corning #3764) using an Echo 550 Liquid Handler (Labcyte). 100 μM benzethonium chloride (BzCl2) and 0.1% dimethyl sulfoxide (DMSO) were used as positive and negative controls respectively. All subsequent liquid handling was performed using MultiFlo FX multi-mode dispenser (BioTek). The pre-dispensed compounds were dissolved in 5 μl of culture media containing cytotoxicity measurement reagents CellTox Green (Promega) and left in plate shaker at RT for 30 min. Twenty microliter cell suspension (75000cells/ ml) was dispensed in the drugged plates. After 72 h incubation, cytotoxicity (fluorescence, 485/520 nm excitation/emission filters) was recorded using PheraStar plate reader (BMG Labtech). Subsequently, 25 μl per well of CellTiter-Glo (Promega) reagent was added, and after 10 min of incubation at room temperature, luminescence (cell viability) was measured using PheraStar plate reader (BMG Labtech).

## Supporting information

**S1 Text. Determining the parameters of the binary model.**
(PDF)

**S1 Appendix. Nonselective effect and drug toxicity.**
(PDF)

**S1 Fig. FIMM drug combination assay example.** 8-by-8 matrices were constructed from Cell Titer-Glo readouts at concentrations 0 nM, 10 nM, 30 nM, 100 nM, 300 nM, 1000 nM, 3000 nM and 10000 nM of each drug compound (vemurafenib and gefitinib in this example). The plotted quantity is the effect $E = -\log(Q)$.
(TIF)

**S2 Fig. Heatmap visualizations of the drug combination screening.** In the in-house combination screen, seven different concentrations in log3-fold dilution ranging from 10 nM to 10000 nM of vemurafenib and gefitinib, vemurafenib and anastrozole, and gefitinib and anastrozole were combined with each other in 8x8 matrix formats in duplicate. For the triple combination, 500 nM of vemurafenib or 500 nM gefitinib or 1000 nM of anastrozole was added to the alternate drug combination matrix (without replicates). In the same way, another set of combination was prepared for vemurafenib, gefitinib and vismodegib. For triple combination, 1000 nM of vismodegib was used whereas the concentrations of vemurafenib and gefitinib were the same as mentioned above.
(TIF)

**S1 Table. Pareto optimal treatments for MALME-3M when only monotherapies and drug duplets are allowed (M = 2).**
(PDF)

**S2 Table. Pareto optimal treatments for MALME-3M when only drug triplets and their subcombinations are allowed (M = 3).**
(PDF)

## Acknowledgments

The authors thank Kaisa Miettinen (University of Jyväskylä) for comments concerning multiobjective optimization, Alexander Ianevski and Anil Kumar (FIMM) for sharing their Excel

spreadsheet of the NCI ALMANAC data, and Laura Turunen (FIMM HTB unit) for the drugs plates in the in-house experiments.

## Author Contributions

**Conceptualization:** Otto I. Pulkkinen, Ville Mustonen, Tero Aittokallio.

**Data curation:** Otto I. Pulkkinen, Prson Gautam.

**Formal analysis:** Otto I. Pulkkinen, Ville Mustonen.

**Funding acquisition:** Ville Mustonen, Tero Aittokallio.

**Investigation:** Otto I. Pulkkinen, Prson Gautam, Ville Mustonen, Tero Aittokallio.

**Methodology:** Otto I. Pulkkinen, Ville Mustonen, Tero Aittokallio.

**Resources:** Ville Mustonen, Tero Aittokallio.

**Software:** Otto I. Pulkkinen.

**Supervision:** Ville Mustonen, Tero Aittokallio.

**Validation:** Prson Gautam.

**Visualization:** Otto I. Pulkkinen.

**Writing – original draft:** Otto I. Pulkkinen, Tero Aittokallio.

**Writing – review & editing:** Otto I. Pulkkinen, Prson Gautam, Ville Mustonen, Tero Aittokallio.

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
