## [Decision Letter · Decision Letter 0]

6 Aug 2020

Dear Dr. Pulkkinen,

Thank you very much for submitting your manuscript "Multiobjective Optimization Identifies Cancer-Selective Combination Therapies" for consideration at PLOS Computational Biology.

As with all papers reviewed by the journal, your manuscript was reviewed by members of the editorial board and by several independent reviewers. In light of the reviews (below this email), we would like to invite the resubmission of a significantly-revised version that takes into account the reviewers' comments.

The reviewers appreciate the importance of optimizing drug combinations and providing greater quantitative insight in this area. However, they raise major concerns regarding the core equations used to establish the drugs' therapeutic and nonselective effects and what the regression model is. Additionally, both reviewers have questions about validating the model predictions.

We cannot make any decision about publication until we have seen the revised manuscript and your response to the reviewers' comments. Your revised manuscript is also likely to be sent to reviewers for further evaluation.

Sincerely,

Stacey Finley, Ph.D.

Associate Editor

PLOS Computational Biology

Jason Papin

Editor-in-Chief

PLOS Computational Biology

Reviewer's Responses to Questions

**Comments to the Authors:**

Reviewer #1: Drug combination is critical for overcoming drug resistance, especially in cancer treatment. In this study, the authors proposed to predict drug combination for specific cancer cells using the multi-objective optimization method. It is a good exploratory design. Some major comments are:

Majors:

Equation (3) have no biological meaning, and would be wrong for most drug combinations. Also it is not a regression model (as there is no parameter, and just simply summarization). Also for the question (4).

Whereas the whole model is built based on this objective function (or minor changes), thus the design of the model might not be reasonable.

Another limitation/concern is the 'pure' data driven, which means a lot of drug combinations should be tested first before the prediction. No new knowledge can be mined from this model (considering the biased/ too simple objective function). Though some predictions are validated, this model is still kind of limited considering the definition of the objective functions (which is unknown for many synergistic or non-effective drug combinations).

Minors:

Drug i at concentration ci is not correct (for each drug there are multiple doses). The same notation error for Ei(ci,cj, l).

Reviewer #2: Reproducibility Report has been uploaded as an attachment.

Reviewer #3: This manuscript explores how non-selective synergistic drug combinations can be identified for a given cancer cell line using NCI-Almanac data. This is posed as a two-objective optimisation problem aiming at identifying combinations that maximise both therapeutic effect and selectivity for the intended cancer cell line. The study focuses on a BRAFV600E-mutant melanoma cell line (MALME-3M), for which they find several optimal solutions (vemurafenib monotherapy and some of its combinations). Two three-drug combinations (triplets) were tested in vitro and the compromises between their non-selectivity and therapeutic effect discussed.

This is definitely an interesting study. I feel however that it has to be polished before publication.

This study employs non-selectivity (essentially which proportion of NCI60 cell lines is inhibited by the drug or combination) as a surrogate of drug safety (toxicity). This assumption is appealing because enables the analysis without requiring further data (e.g. activity of the drug/s on a non-tumoural cell line). Ideally, this plausible hypothesis should be supported by some form of validation. Looking at adverse effects arising from clinical practice (drugs.com) and Figure 1C, NCI60-selective vemurafenib has a higher number of important adverse effects than NCI60-promiscuous mythramycin, and thus the assumption does not seem to hold in this case. The other aspect to discuss is related to intra-tumour heterogeneity, which is minimal in cancer cell lines (one clone at low passages) but typically high in primary tumours (clones with different drug sensitivies). If we see each melanoma cell line as a tumour clone, it is not unreasonable to think that in vitro non-selective drugs might be able to delay the emergence of acquired resistance more than in vitro selected drugs due to the former neutralising a larger proportion of the tumour clones. In that case, non-selectivity would be positive rather than negative. I think that the study would benefit from discussing the limitations of this assumption.

In Figure 1, all 104 NCI-Almanac are being used, which come from three screening centres (mostly FG and FF). For a given drug-cell line pair, the single-drug activities from FF exhibit much higher variability than those from FG (https://doi.org/10.3389/fchem.2019.00509). Were all single-drug activities for a given cell line considered in calculating Qmin? If not, how were extreme values discarded? The authors should discuss how Figure 1C changes depending on the level of variability in single-drug activities.

On the other hand, pages 3 to 6 (up to ‘Application to melanoma’) present a sound methodological development, but should be made clearer and better organised. Adding subsections such as Monotherapy, Two-drug Combinations and Higher-order Combinations would be helpful, each with their specific therapeutic effect and non-selectivity equations.

The two-objective optimization problem is solved for monotherapies and two-drug combinations by inspecting NCI-Almanac data (a reference to the epsilon-constrained method is required). It is not clear either how this is done for the three-drug combinations, as such data is not available at NCI-Almanac. In page 3, a regression model is mentioned, but which quantity is predicted and how? (algorithm, data, features). Also, I am missing the definition of M in equation 6. M=2 seem to cover monotherapies and two-drug combinations, that is N=1 and N=2 in equation 3, but why?

Page 7 (last paragraph) – very interesting result to discuss: vemurafenib + gefitinib’s synergy depends on their concentration to the extent of being antagonistic for some concentration pairs. Could the authors discuss which of these in vitro concentrations would be relevant in vivo? That is, any way to anticipate whether synergy or antagonism in vivo given these in vitro results?

The conclusions are summarised in the last paragraph of the discussion. I am missing a summary of which two-objective-optimal monotherapies, two-drug combinations and higher-order combinations have been identified in NCI-Almanac and, in the latter case, confirmed in vitro by the authors.

**Have all data underlying the figures and results presented in the manuscript been provided?**

Reviewer #1: None

Reviewer #2: None

Reviewer #3: Yes

PLOS authors have the option to publish the peer review history of their article (what does this mean?). If published, this will include your full peer review and any attached files.

Reviewer #1: No

Reviewer #2: **Yes: **Anand K Rampadarath

Reviewer #3: **Yes: **Pedro J. Ballester
---

## [Decision Letter · Decision Letter 1]

28 Oct 2020

Dear Dr. Pulkkinen,

Thank you very much for submitting your manuscript "Multiobjective Optimization Identifies Cancer-Selective Combination Therapies" for consideration at PLOS Computational Biology. As with all papers reviewed by the journal, your manuscript was reviewed by members of the editorial board and by several independent reviewers. The reviewers appreciated the attention to an important topic. Based on the reviews, we are likely to accept this manuscript for publication, providing that you modify the manuscript according to the review recommendations.

Overall, the reviewers agree that the updated manuscript is much improved. One issue remains, related to the limitations of the work, including the model's ability to predict new combination treatments. This can be addressed in a revised manuscript.

Sincerely,

Stacey Finley, Ph.D.

Associate Editor

PLOS Computational Biology

Jason Papin

Editor-in-Chief

PLOS Computational Biology

[LINK]

Reviewer's Responses to Questions

**Comments to the Authors:**

Reviewer #1: The revision adressed the comments, and is acceptable with minor revisions.

Minor comments:

1) it is better to discuss the potential limitations mentioned in the review.

2) It is better to discuss whether the model can predict new combinations (without combination experimental data), or select the 'best' combinations based on the screening data of drug combinations.

Reviewer #3: The authors have done an excellent job at addressing all my comments and I agree that the paper has improved very much as a result. I do not have any further comment.

**Have all data underlying the figures and results presented in the manuscript been provided?**

Reviewer #1: None

Reviewer #3: Yes

PLOS authors have the option to publish the peer review history of their article (what does this mean?). If published, this will include your full peer review and any attached files.

Reviewer #1: No

Reviewer #3: **Yes: **Pedro J. Ballester
---

## [Editor Report · Decision Letter 2]

13 Nov 2020

Dear Dr. Pulkkinen,

We are pleased to inform you that your manuscript 'Multiobjective Optimization Identifies Cancer-Selective Combination Therapies' has been provisionally accepted for publication in PLOS Computational Biology.

Best regards,

Stacey Finley, Ph.D.

Associate Editor

PLOS Computational Biology

Jason Papin

Editor-in-Chief

PLOS Computational Biology

---

## [Editor Report · Acceptance letter]

21 Dec 2020

PCOMPBIOL-D-20-00963R2 

Multiobjective Optimization Identifies Cancer-Selective Combination Therapies

Dear Dr Pulkkinen,

I am pleased to inform you that your manuscript has been formally accepted for publication in PLOS Computational Biology. Your manuscript is now with our production department and you will be notified of the publication date in due course.

With kind regards,

Jutka Oroszlan
